

# Genome-wide identification and expression analysis of calmodulin-like proteins in cucumber

Yunfen Liu[1,2], Feilong Yin[1], Lingyan Liao[1] and Liang Shuai[1,2]

[1] College of Food and Biological Engineering/Institute of Food Science and Engineering Techology, Hezhou University, Hezhou, Guangxi, China
[2] Guangxi Key Laboratory of Health Care Food Science and Technology, Hezhou University, Hezhou, Guangxi, China

## ABSTRACT

**Background**. The calmodulin-like (CML) protein is a crucial $Ca^{2+}$-binding protein that can sense and conduct the $Ca^{2+}$ signal in response to extracellular stimuli. The CML protein families have been identified and characterized in many species. Nevertheless, scarce information on cucumber CML is retrievable.

**Methods**. In this study, bioinformatic analyses, including gene structure, conserved domain, phylogenetic relationship, chromosome distribution, and gene synteny, were comprehensively performed to identify and characterize *CsCML* gene members. Spatiotemporal expression analysis in different organs and environment conditions were assayed with real-time quantitative polymerase chain reaction (qRT-PCR).

**Results**. Forty-four CsCMLs family members were well characterized, and the results showed that the 44 CsCML proteins contained one to four EF-hand domains without other functional domains. Most of the CsCML proteins were intron-less and unevenly distributed on seven chromosomes; two tandemly duplicated gene pairs and three segmentally duplicated gene pairs were identified in the cucumber genome. *Cis*-acting element analysis showed that the hormone, stress, and plant growth and development-related elements were in the promotor regions. In addition, spatiotemporal expression analysis revealed distinctive expression patterns for *CsCML* genes in different tissues and environmental conditions, and a putative protein interaction network also confirmed their potential role in responding to various stimuli. These results provide a foundation for understanding *CsCML*s and provide a theoretical basis for further study of the physiological functions of *CsCML*s.

## INTRODUCTION

Calcium ions ($Ca^{2+}$) are essential nutrients in plant growth and development and are also the primary second messengers of all eukaryotic cells in cell signal transduction (*Zeng et al., 2017*). When plants suffer stress from external stimuli (salt, temperature, drought, heavy metal toxicity, light, plant hormones, and pathogenic microorganism infection, among others), changes in the concentration of intracellular and extracellular $Ca^{2+}$ occur, evoking

Corresponding author
Liang Shuai, shuailiang1212@163.com

a calcium signature (*Perochon et al., 2011*). The downstream conduction of calcium signals requires the involvement of calcium-binding proteins, which are referred to as calcium sensors. The major sensors include calmodulins (CaMs), CaM-like proteins (CMLs), $Ca^{2+}$-dependent protein kinases (CDPKs), and calcineurin B-like proteins (CBLs) (*Ranty et al., 2016*; *Yang & Poovaiah, 2003*). They commonly contain elongation factor hand (EF-hand) motifs. The EF-hand is a helix-loop-helix structure capable of binding $Ca^{2+}$, following which it undergoes a conformational change, interacting with downstream proteins or modulating its own catalytic activity (*Kim et al., 2009*; *Perochon et al., 2011*; *Zeng et al., 2017*). The CML is a $Ca^{2+}$ sensor protein similar to CaM and contains several EF-hands that bind to $Ca^{2+}$ andact on downstream targets (*Mohanta, Kumar & Bae, 2017*). These sensors can be divided into two groups: sensor relays, such as CaM, CML, and CBL, which do not have catalytic activity themselves and need to interact with downstream target proteins and form similar $Ca^{2+}$/CaM complexes; and sensor responders, which contain other effector domains except for EF-hands, directly relaying the signal to downstream targets, such as CDPKs (*Perochon et al., 2011*).

As a typical $Ca^{2+}$ sensor in higher plants, CMLs possess one to six EF-hand motifs without any other functional domain (*McCormack, Tsai & Braam, 2005*). To date, *CML* genes have been analyzed in various species; such as, 50 have been identified in *Arabidopsis* (*McCormack & Braam, 2003*), 32 in rice (*Boonburapong & Buaboocha, 2007*), 68 in grape (*Konstantin et al., 2021*), 79 in Chinese cabbage (*Nie, Zhang & Zhang, 2017*), 52 in tomato (*Munir et al., 2016a*), 50 in alfalfa (*Sun, Yu & Guo, 2020*), 21 in ginkgo (*Zhang et al., 2022*), 82 in chrysanthemum (*Fu et al., 2022*), 41 in soybean (*Yadav et al., 2022*), and 58 in apple (*Li et al., 2019a*; *Li et al., 2019b*). Although the cucumber genome has been well assembled and annotated (*Cavagnaro et al., 2010*; *Li et al., 2019a*; *Li et al., 2019b*; *Osipowski et al., 2020*), the characteristics and functions of *CML* gene members are still unknown. Previous research showed that CMLs play a critical role in plant growth and development, stress, and hormone responses. In *Arabidopsis*, *CML23* and *CML24* are associated with plant flowering (*Tsai et al., 2007*); *AtCML4/5* functions in vesicle transport within the plant endomembrane system (*Ruge et al., 2016*); and promoter analysis showed that *AtCML15* and *AtCML16* might have a role in floral development (*Ogunrinde et al., 2017*). *AtCML8* (*Zhu et al., 2017*) and *AtCML9* (*Leba et al., 2012*) were verified to participate in the plant immune response against *Pseudomonas syringae*; *AtCML37* and *AtCML38* were induced by wounding, osmotic stress, and drought; while *AtCML39* was dramatically expressed when stimulated by methyl jasmonate (MeJA) (*Vanderbeld & Snedden, 2007*). In soybean, *GsCML27* was induced by salt, bicarbonate, and osmotic stress, and the ectopic expression of *GsCML27* decreased by changing the content of cell ions and osmotic regulation (*Chen et al., 2015*). In *Vitis amurensis*, *CML41b*, *CML71*, *CML54*, and *CML85* were induced by UV-C and plant hormones, such as MeJA and salicylic acid (SA) (*Konstantin et al., 2021*). The CML family members thus have diverse functions in plant development and stress resistance.

As sequencing technology has become increasingly accessible, the whole-genome identification of functional genes is no longer limited to model plants. Cucumber (*Cucumis sativus* L.) is an economically important vegetable worldwide (*Ali, Maryam & Seyed, 2016*).

As the cucumber genome sequence has been completed (*Huang et al., 2009*), the *CML* gene family can be comprehensively analyzed and characterized. Although several literatures have reported the certain CsCMLs, to date, systematical analysis of cucumber CML have not been conducted. The aim of this study was thus to identify the putative *CML* family members in the cucumber genome and analyze the structure, evolutionary relationships, chromosomal distribution, and promoter elements. Furthermore, the expression patterns of the *CsCML*s were also detected under phytohormone and abiotic stress treatments in different organs. The findings of this study provide a foundation for understanding the functional characteristics of *CsCML*s at the physiological and molecular levels.

## MATERIAL AND METHODS

### Genome-scale identification of CML in cucumber

To identify CML family members in cucumber, the cucumber, Arabidopsis, and tomato genome database were downloaded from EnsemblPlant (*Yates et al., 2022*) (http://jul2018-plants.ensembl.org/index.html), and for *Arabidopsis*, 50 CML proteins were retrieved from UniProt (https://www.uniprot.org/) (*McCormack & Braam, 2003*); additionally, 32 rice CML proteins were downloaded from TIGR (http://rice.plantbiology.msu.edu/) (the protein sequences were shown in Data S1). Fifty CML protein sequences of *Arabidopsis* were used as queries to search against the cucumber peptides for the first blast by TBtools. BlastP was used for the second blast to obtain candidate cucumber CMLs with an e-value lower than $10^{-5}$, and the redundant and repetitive sequences were removed manually. NCBI-CDD (http://www.ncbi.nlm.nih.gov/Structure/cdd/wrpsb.cgi) and Interproscan (http://www.ebi.ac.uk/interpro/search/sequence/), and SMART (http://smart.embl-heidelberg.de) were used to predict EF-hand domains, eliminating the protein sequences without EF-hands or with other functional domains. And blasted with AtCaM2(Accession number: NP_850344.1) which acted as typical CaMs (*McCormack & Braam, 2003*) as well as the amino acid identity less than 80% to ensured the CMLs. The identified genes were named *CsCML1* to *CsCML44*, and the nucleotide and putative amino acid sequences were used for further analysis.

### Sequence analysis

The physicochemical parameters of CsCMLs, including the molecular weight (MW), theoretical point (pI), instability index, grand average of hydropathicity (GRAVY), aliphatic index, and number of amino acids, were predicted using ExPASyProtParam (http://web.expasy.org/protparm/). N-terminal myristoylation and S-palmitoylation were analyzed by GSP-Lipid (http://lipid.biocuckoo.org/webserver.php), and the subcellular location was predicted by Wolf PSORT (http://www.genscript.com/psort/wolf_psort.html). To predict the number of EF-hands, SMART (http://smart.embl-heidelberg.de/smart/set_mode.cgi?NORMAL=1) was used.

### Structure analysis and phylogenetic tree construction of CsCMLs

MEME (http://meme-suite.org/index.html) was used to analyze the conserved domains, and the number of motif was set to 6. The exon-intron structure was analyzed by GSDS

(http://gsds.gao-lab.org/). The *CsCML* nucleotide sequences were retrieved from the cucumber genome. Two-kilobase upstream sequences were considered to be promoters, and PlantCARE (http://bioinformatics.psb.ugent.be/webtools/plantcare/html/) (*Lescot et al., 2006*) was used to analyze the *cis*-acting elements in the promoter region, and visualized by Simple BioSequence Viewer in TBtools (*Chen et al., 2020*). The phylogenetic tree was constructed using the neighbor-joining method in MEGA7 (*Kumar, Stecher & Tamura, 2016*) with 1000 bootstrap replicates. Classification of the CsCML proteins was performed based on the phylogenetic relationships with 50 Arabidopsis AtCMLs, 32 rice OsCMLs.

## Chromosomal distribution and syntenic analysis

The *CsCML* genes were mapped to the cucumber genome database based on physical location information. *CsCML* gene duplication was analyzed according to Multiple Collinearity ScanX (MCSanX). Synteny was analyzed among *Arabidopsis* and tomato using TBtools (*Sun et al., 2018*).

## Plant materials and treatments

Cucumber ('Fengshou 3 hao') seeds were germinated on two layers of moist gauze in a light incubator (RXZ type, Ningbo, China) at 28 °C for 24 h. The germinated seeds were transplanted into soil (Jiffy substrates, Jiffy International AS, Kristiansand, Netherlands) under a 16 h light/8 h dark cycle and 85–90% humidity. Three-week-old seedlings were used to assay the effects of phytohormone and abiotic stress treatments (low temperature and drought). For phytohormone analysis, the seedlings were sprayed with ABA (abscisic acid, 100 $\mu$mol/L) or GA$_3$ (gibberellic acid, 100 $\mu$mol/L). For low-temperature stress, the seedlings were placed at 5 °C for 3 h in an incubator; for drought stress, the seedlings were removed from the soil, the soil was removed from the seedlings, and then the seedlings were left at room temperature for 3 h (*Munir et al., 2016a*; *Munir et al., 2016b*). Seedlings without any treatment were used as control. All treatments were performed using three biological replicates, and two mature leaves were collected, immediately frozen in liquid nitrogen, and stored at −80 °C for further analysis.

## RNA extraction and gene expression pattern analysis

Total RNA was extracted from the leaves, stems, flowers, and peels. Frozen samples were well ground to powder in liquid nitrogen before extracting according to the manufacturer's instructions of the RNA extraction kit (Sangon Biotech, Shanghai, China). RNA integrity was electrophoresed on 1% (w/v) agarose gel, and then RNA was quantified using a micro spectrophotometer (KAIAO, Guongdong, China). Total RNA (1 $\mu$g) was used for cDNA synthesis using the 5X All-In-One RT MasterMix (Abm, Richmond, BC, Canada). Real-time quantitative reverse transcription-polymerase chain reaction (qRT-PCR) was performed using EvaGreen 2X qPCR MasterMix-No Dye (Abm, Richmond, BC, Canada) with a fluorescence qPCR instrument (BioRad, Hercules, CA, USA). The specific primers used for qRT-PCR are listed in Table S1. The cucumber *Actin* gene (accession number: XM_011659465.2) was used as an internal control. Real-time PCR was executed for triplicates. Relative expression was analyzed using the $2^{-\Delta\Delta Ct}$ method (*Livak & Schmittgen, 2001*). The data expressed represent the average of three biological replicates.
## Protein interaction network prediction

The 44 CsCML protein sequences were submitted to the online server STRING (https://cn.string-db.org/cgi/input?sessionId=bvfXM8z80Gzw{&}input_page_show_search=on, version 11.5).

## Statistical analysis

Each experiment contained three independent biological replicates. The gene expression assayed was conducted three technical replicates and the data was processed by Excel and represented as the mean ± standard error (SE).

# RESULTS

## Genome-wide identification and characterization of CML in cucumber

As a result, 44 putative CML family members in cucumber were obtained and subjected to Pfam, InterProScan, and SMART to verify the EF-hand conserved domain. The gene name, gene ID, number of amino acids, amono acids identity to AtCaM2, MW, pI, number of EF-hand domains, GRAVY, predicted subcellular location, N-terminal myristoylation, S-palmitoylation, instability index, and aliphatic index were listed in Table 1. The number of amino acids in the CsCML proteins ranged from 81 to 251. The CsCMLs shared 24%–77% identity with AtCaM2. The MW of CsCML1–44 varied from 9.187 (CsCML20) to 26.355 kDa (CsCML16), and the pI ranged from 3.78 (CsCML11) to 9.17 (CsCML32). Most CsCML proteins contained two to four EF-hand domains, except CsCML4, which possessed only one.

Moreover, most CsCMLs were cytosolic and nuclear proteins, but some were plastid ones, such as chloroplast and mitochondrial proteins. The GRAVY values of most CsCML proteins were negative, indicating that CML proteins in cucumber are hydrophilic. The analysis of myristoylation and palmitoylation indicated that only CsCML15, CsCML16, CsCML28, and CsCML32 had N-terminal myristoylation sites, and eight CsCMLs had S-palmitoylate sites, which implies that these CsCML proteins may have membrane-protein interactions. Half of the CML proteins were unstable.

## Structure analysis, conserved motif and *cis*-acting elements in promotor of CMLs in cucumber

To further investigate the features of the CsCML proteins, conserved motifs were identified using the MEME program, and six distinct motifs were identified (Fig. 1A). Motifs 1 and 2 were present in all 44 CsCML family members, except CsCML44 and CsCML28, which lacked motif 2. Some paralogous proteins contained different motifs, such as CsCML6/21, CsCML11/20, CsCML1/30, CsCML33/34, and CsCML3/4, while CsCML5/8, CsCML14/17, CsCML19/42, CsCML7/41, and CsCML31/32 shared a similar motif. Motif 6 was only found in CsCML3/4. These results might indicate that these paralogous CsCMLs are more diverse than the nonhomologous proteins. Therefore, further analysis should explore the function of the CsCML protein members.

Exon-intron analysis showed that 35 of the total CsCML members had no introns, while 9 had one to six introns (only CsCML29 had six introns; Fig. 1B). Intron phases

Liu et al. (2023), *PeerJ*, DOI 10.7717/peerj.14637

**Table 1   Information on *CsCML*s in cucumber.**

| Gene name | Gene ID | Number of amino acids | % of amono acids identity to AtCaM2 | Molecular weight (KDa) | pI | Number of EF-hand domain | Predicted subcellular location | GRAVY | N-terminal myristoylation site | S-Palmitoylation site | Instability index | Aliphatic index |
|---|---|---|---|---|---|---|---|---|---|---|---|---|
| *CsCML1* | KGN43936 | 184 | 33 | 21.06 | 8.83 | 4 | nucl: 7, cyto: 3, mito: 3 | −0.666 | | | unstable | 68.32 |
| *CsCML2* | KGN46235 | 178 | 38 | 20.92 | 5.5 | 4 | chlo: 8, nucl: 3, extr: 2 | −0.560 | | | unstable | 70.11 |
| *CsCML3* | KGN47684 | 165 | 28 | 19.03 | 4.22 | 3 | cyto: 9, chlo: 4 | −0.312 | | + | unstable | 85.64 |
| *CsCML4* | KGN50423 | 174 | 33 | 20.63 | 4.54 | 1 | cyto: 7, nucl: 5, chlo: 1 | −0.630 | | | unstable | 75.57 |
| *CsCML5* | KGN51467 | 182 | 46 | 19.71 | 4.34 | 4 | chlo: 5, mito: 5, nucl: 3 | −0.388 | | | stable | 75.49 |
| *CsCML6* | KGN51513 | 145 | 31 | 15.90 | 4.34 | 4 | cyto: 6, chlo: 4, extr: 3 | −0.079 | | | unstable | 89.38 |
| *CsCML7* | KGN52027 | 161 | 41 | 17.74 | 4.36 | 4 | nucl: 6, cyto: 5.5, nucl_plas: 4.5, cyto_E.R.: 3.5 | −0.234 | | | stable | 84.29 |
| *CsCML8* | KGN52351 | 185 | 48 | 20.12 | 4.33 | 4 | chlo: 7, mito: 5, nucl: 2 | −0.332 | | | unstable | 77.95 |
| *CsCML9* | KGN52470 | 180 | 46 | 20.07 | 4.5 | 2 | nucl: 5, chlo: 4, extr: 4 | −0.168 | | | unstable | 76.83 |
| *CsCML10* | KGN54524 | 192 | 37 | 21.26 | 4.85 | 3 | mito: 6, nucl: 4, cyto: 3 | −0.566 | | | unstable | 58.39 |
| *CsCML11* | KGN54599 | 157 | 35 | 16.82 | 3.78 | 2 | cyto: 9.5, cyto_ER: 5.5, chlo: 1, mito: 1, plas: 1 | 0.138 | | | stable | 102.55 |
| *CsCML12* | KGN55185 | 90 | 32 | 10.62 | 9 | 2 | mito: 12, chlo: 1 | −0.768 | | | stable | 76 |
| *CsCML13* | KGN55186 | 97 | 29 | 11.00 | 5.29 | 2 | chlo: 9, nucl: 3, cyto: 1 | −0.375 | | | stable | 83.51 |
| *CsCML14* | KGN55728 | 163 | 46 | 18.12 | 4.48 | 4 | nucl_plas: 5.5, plas: 5, nucl: 4, cyto: 2, mito: 2 | −0.361 | | | stable | 83.31 |
| *CsCML15* | KGN55913 | 147 | 48 | 16.73 | 4.82 | 3 | cyto: 7, plas: 3, nucl: 2, extr: 1 | −0.526 | + | | stable | 78.3 |
| *CsCML16* | KGN56046 | 229 | 28 | 26.36 | 4.52 | 3 | cyto: 7, nucl: 3, chlo: 2, extr: 1 | −0.359 | + | + | unstable | 78.3 |
| *CsCML17* | KGN56734 | 227 | 25 | 26.05 | 5.51 | 4 | chlo: 8, cyto: 3, extr: 2 | −0.482 | | + | unstable | 78.15 |
| *CsCML18* | KGN56931 | 150 | 77 | 17.06 | 4.08 | 4 | cyto: 6, chlo: 3, extr: 2, cysk: 1.5, cysk_plas: 1.5 | −0.381 | | | stable | 84.47 |
| *CsCML19* | KGN57816 | 156 | 41 | 17.01 | 4.62 | 4 | chlo: 6, extr: 6, nucl: 1 | −0.397 | | + | stable | 78.14 |
| *CsCML20* | KGN57835 | 83 | 38 | 9.19 | 4.38 | 2 | nucl: 11, mito: 3 | −0.496 | | | stable | 57.83 |
| *CsCML21* | KGN58193 | 141 | 34 | 15.94 | 4.72 | 4 | nucl: 7, chlo: 4, cyto: 1, mito: 1 | −0.495 | | + | unstable | 70.5 |
| *CsCML22* | KGN58678 | 201 | 33 | 22.72 | 4.42 | 3 | nucl: 9, nucl_plas: 6.5, plas: 2, chlo: 1 | −0.518 | | | unstable | 76.12 |
| *CsCML23* | KGN59507 | 167 | 29 | 18.27 | 4.77 | 2 | mito: 7, nucl: 4, chlo: 3 | −0.375 | | | unstable | 65.45 |
| *CsCML24* | KGN59556 | 210 | 40 | 24.51 | 4.79 | 2 | nucl: 10, cyto: 3 | −0.285 | | | stable | 86.29 |
| *CsCML25* | KGN59929 | 180 | 36 | 20.39 | 4.86 | 3 | chlo: 8, mito: 2, vacu: 2, cyto: 1 | −0.398 | | | unstable | 67.17 |
| *CsCML26* | KGN60706 | 227 | 45 | 25.47 | 4.59 | 4 | chlo: 11, cyto: 2 | −0.247 | | | unstable | 73.79 |
| *CsCML27* | KGN60815 | 146 | 64 | 16.84 | 4.07 | 4 | cyto: 12, cysk: 1 | −0.390 | | | unstable | 85.48 |
| *CsCML28* | KGN61657 | 162 | 60 | 18.79 | 4.53 | 4 | nucl: 5, cyto: 4, pero: 2, plas: 1, extr: 1 | −0.686 | + | | stable | 72.22 |
| *CsCML29* | KGN62011 | 168 | 46 | 19.23 | 4.72 | 4 | cyto: 6, nucl: 3, chlo: 2, extr: 2 | −0.860 | | | stable | 63.33 |
| *CsCML30* | KGN62012 | 190 | 38 | 21.16 | 5.59 | 4 | mito: 7, cyto: 4, chlo: 3 | −0.578 | | + | stable | 64.11 |
| *CsCML31* | KGN62512 | 87 | 30 | 9.81 | 5.85 | 2 | cyto: 7, mito: 4, chlo: 3 | −0.316 | | | stable | 66.21 |
| *CsCML32* | KGN62513 | 81 | 33 | 9.42 | 9.17 | 2 | cyto: 11, nucl: 2 | −0.536 | + | + | stable | 87.9 |
| *CsCML33* | KGN62517 | 137 | 29 | 15.45 | 4.88 | 3 | mito: 5, cyto: 4, chlo: 2, nucl: 2 | −0.499 | | | unstable | 82.63 |
| *CsCML34* | KGN62522 | 100 | 31 | 11.68 | 5.16 | 2 | chlo: 6, extr: 3, nucl: 2, cyto: 2 | −0.658 | | + | stable | 70.1 |
| *CsCML35* | KGN62524 | 161 | 26 | 18.13 | 5.39 | 3 | extr: 6, nucl: 4, cyto: 2, chlo: 1 | −0.681 | | | unstable | 76.4 |
| *CsCML36* | KGN62525 | 87 | 35 | 10.11 | 5.09 | 2 | cyto: 9, nucl: 2, mito: 1, extr: 1 | −0.569 | | | unstable | 71.84 |

Liu et al. (2023), *PeerJ*, DOI 10.7717/peerj.14637

**Table 1** (*continued*)

| Gene name | Gene ID | Number of amino acids | % of amono acids identity to AtCaM2 | Molecular weight (KDa) | pI | Number of EF-hand domain | Predicted subcellular location | GRAVY | N-terminal myristoylation site | S-Palmitoylation site | Instability index | Aliphatic index |
|---|---|---|---|---|---|---|---|---|---|---|---|---|
| *CsCML37* | KGN62526 | 171 | 23 | 19.19 | 9.11 | 3 | nucl: 7.5, nucl_plas: 4.5, cyto: 3, chlo: 1, mito: 1 | −0.651 | | | unstable | 79.88 |
| *CsCML38* | KGN62844 | 174 | 38 | 19.33 | 4.58 | 2 | cyto: 9, nucl: 2, chlo: 1, mito: 1 | −0.128 | | | stable | 96.84 |
| *CsCML39* | KGN63180 | 140 | 31 | 15.66 | 4.37 | 4 | cyto: 6, chlo: 4, nucl: 3 | −0.325 | | | stable | 78.57 |
| *CsCML40* | KGN63491 | 142 | 33 | 15.88 | 4.51 | 3 | cyto: 6, chlo: 3, mito: 3, nucl: 1 | −0.409 | | | unstable | 87.25 |
| *CsCML41* | KGN63786 | 160 | 41 | 17.37 | 4.11 | 4 | chlo: 7, nucl_plas: 3.5, plas: 3, nucl: 2, cyto: 1 | 0.059 | | | stable | 93.37 |
| *CsCML42* | KGN63870 | 188 | 39 | 20.57 | 4.33 | 4 | nucl: 4, nucl_plas: 4, chlo: 3, mito: 3, cyto: 2, plas: 2 | −0.562 | | | stable | 76.91 |
| *CsCML43* | KGN64007 | 204 | 42 | 23.39 | 4.34 | 2 | cyto: 6, ER: 2, cysk: 2, golg: 2, nucl: 1 | −0.070 | | | unstable | 91.67 |
| *CsCML44* | KGN64933 | 251 | 24 | 28.31 | 5.26 | 2 | nucl: 9.5, cyto_nucl: 5.5, chlo: 3 | −0.550 | | | unstable | 58.29 |

**Notes.**

Cyto, cytosol; ER, endoplasmic reticulum; Vacu, vacuolar; membrane; Chlo, chloroplast; Nucl, nuclear; Extr, extracellular; Mito, mitochondria; Cysk, cytoskeleton; Plas, plasmamembrane; MW, molecular weight; pI, theoretical isoelectric point of proteins; GRAVY, grand average of hydropathicit.

+ means presence.

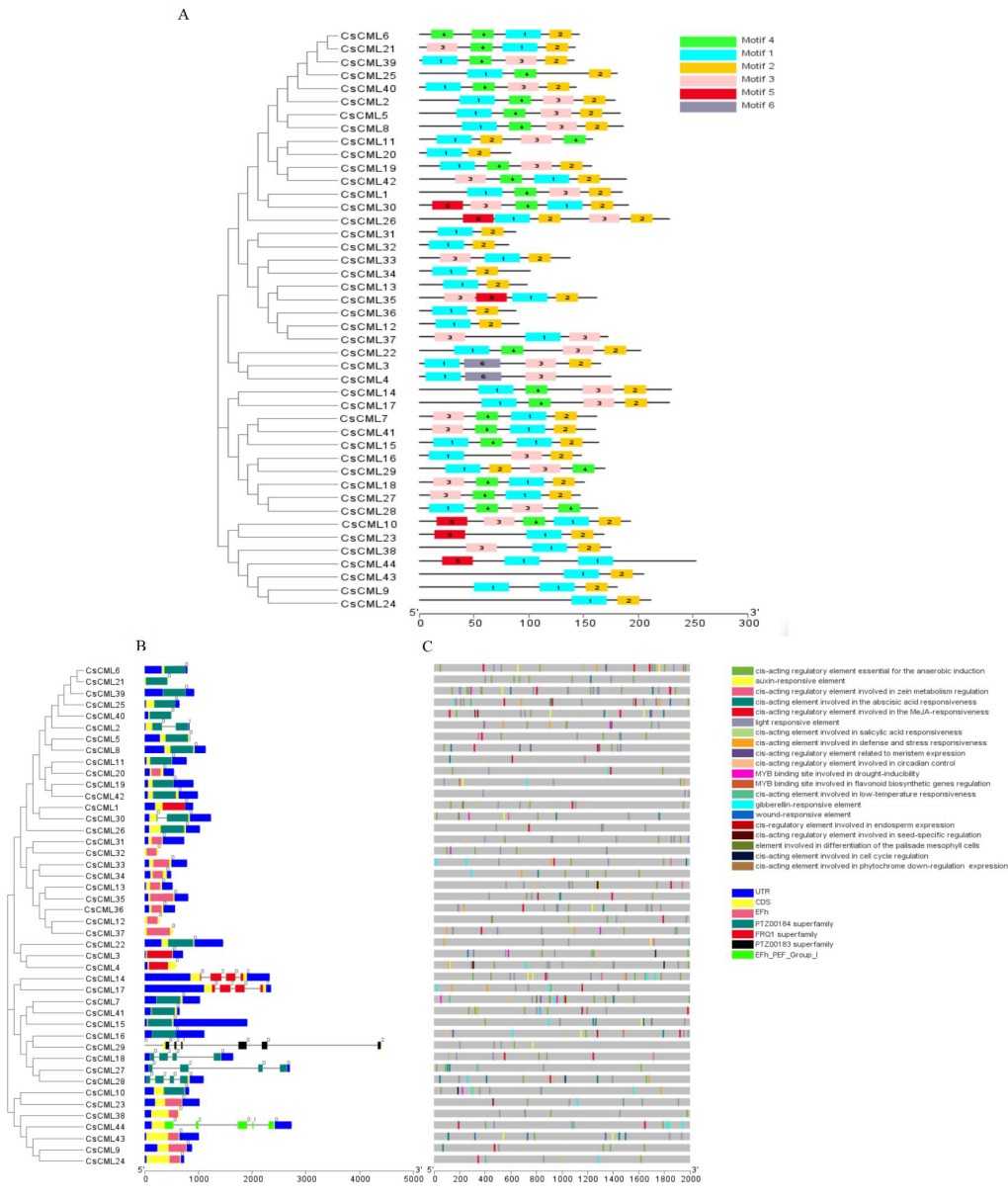

**Figure 1  Phylogenetic relationship, gene structure, conserved protein motifs, and putative *cis*-acting elements in *CsCMLs*.** Phylogenetic relationship, gene structure, conserved protein motifs, and putative cis-acting elements in CsCMLs. (A) Distribution of the motifs in the CsCML proteins. Motifs 1–6 are displayed with different colored boxes. (B) Exon-intron structure and EF-hand domain. The grey dashed line represents the intron. (C) Putative cis-acting elements.

concerning codons were also investigated. The numbers 0, 1, and 2 indicated that splicing occurred after the first, second, and third nucleotide in the codon, respectively. Most of the first introns were phase 0 introns, suggesting that the splicing phase is highly conserved in cucumber. The CsCML proteins contained five conserved EF-hand (Fig. 1B). Most of

them belonged to EF-hand domain and the PTZ00184 superfamily. Only CsCML29 and CsCML44 belonged to PTZ00183 superfamily and EFh_PEF_Group 1, respectively.

To better understand the transcriptional regulation of *CsCMLs*, the *cis*-acting elements were investigated in the upstream 2000-bp sequences for *CsCML* s using PlantCARE2.0 (Fig. 1C, Data S2). The major elements were related to plant hormone responsiveness, light responsiveness, defense and stress responsiveness, and plant growth and development, typically including circadian control, meristem expression, and seed-specific regulation. Figure 2 shows the number of *CsCMLs* containing *cis*-acting elements. All *CsCML* gene promoter regions contained G-box/GT1-motif, which is related to light responsiveness. Of the *CsCML* gene promoter regions, 32 contained the ABRE motif, which is the *cis*-acting element involved in ABA responsiveness. Moreover, the promoter region of the *CsCML* contained MeJA (CGTCA-motif, TGACG-motif), SA (TCA element), GA (P-box and GARE-motif), and auxin (TGA element) responsiveness elements. Of the *CsCML* gene promoter regions, 40 contained the anaerobic induction regulatory element. The TC-rich repeats involved in defense and stress responsiveness were present in 17 *CsCML* genes. Additional stress responsiveness elements, such as the wound-responsive element WUN-motif, drought-inducibility element MBS, and low-temperature response element LTR, were also found. Some motifs involved in plant growth and development, such as circadian control, meristem expression (CAT-box), seed-specific regulation (RY-element), and endosperm expression (GCN4-motif), were observed in a few genes. Besides, some CsCMLs promoter regions contained the element which involved in zein metabolism regulation and flavonoid biosynthetic genes regulation. Overall, the analysis of *cis*-acting elements suggested that the family members of *CsCML* genes play different and complex roles in plant growth and development and stress responsiveness.

## Phylogenetic relationships of CML proteins in cucumber, Arabidopsis and rice

To explore the the systematic evolution of CsCMLs, we constructed the phylogenetic tree with 44 CsCMLs, 50 AtCMLs, and 32 OsCMLs (Fig. 3). The tree showed that the CML were classified into eight subgroups (I–VIII) based on the classification of AtCMLs, and most CMLs were in subgroups II (23 CMLs) and VI (24 CMLs). The smallest subgroup was V which consisted of six CMLs while without any OsCML. Notably, in subgroup I, several CsCMLs monopolized a small branch, which might result in the special function of these CsCMLs.

## Chromosomal location and duplication of CsCML and synteny analysis in cucumber, *Arabidopsis thaliana*, and tomato

The identified *CsCML* genes were mapped to the seven chromosomes of the cucumber genome database. The results showed that 44 *CsCML* family members were diversely spread across all seven chromosomes (Fig. 4A). The highest number of genes (14) was located on chromosome 2, followed by chromosome 3 with 12 genes. Chromosomes 1, 4, and 5 contained five, four, and six *CsCML* genes, respectively. However, there were only one and two genes located on chromosomes 7 and 6, respectively.

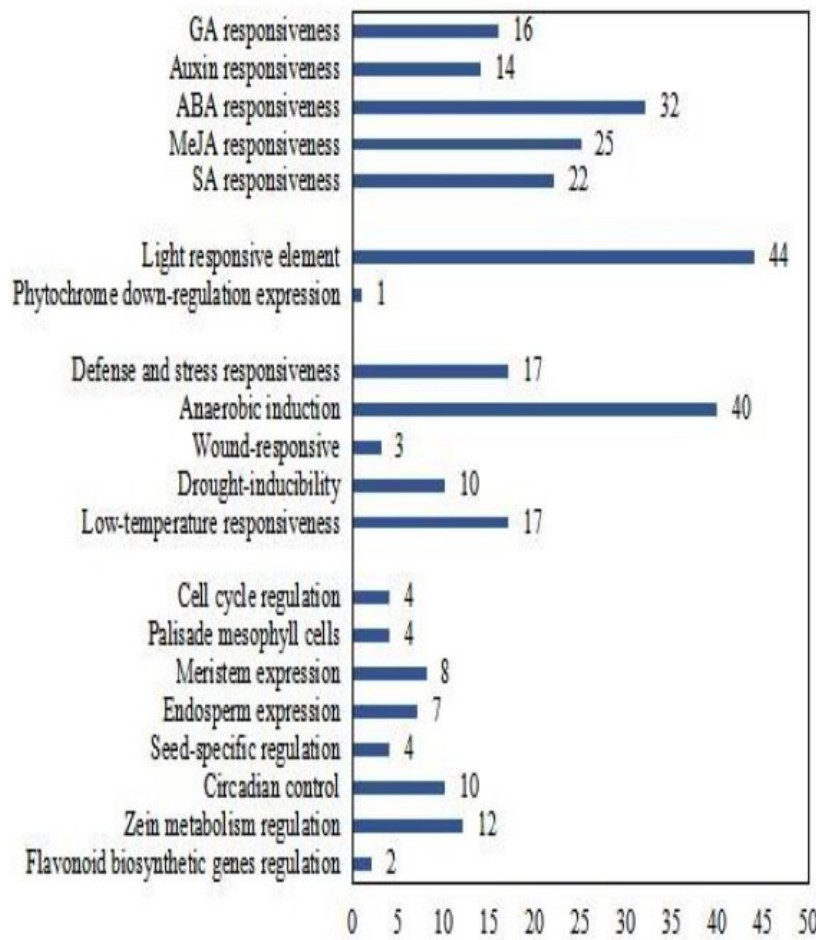

**Figure 2  Number of *CsCML* genes containing *cis*-acting elements.**

To determine the gene duplication of the 44 *CsCML*s, segmental duplication was analyzed using TBtools (Figs. 4A and 4B). Two tandemly duplicated gene pairs *CsCML31/CsCML32* and *CsCML35/CsCML36* (Fig. 4A) and three segmentally duplicated gene pairs *CsCML28/CsCML18*, *CsCML35/CsCML13*, and *CsCML37/CsCML12* (Fig. 4B) were identified in the cucumber genome. To further understand the evolution of *CML* genes, collinear analysis related to the *Arabidopsis,* tomato, and cucumber genomes was performed (Fig. 4C Data S4). Collinear gene pairs were present between cucumber and both *Arabidopsis* and tomato. There were many more *CsCML*syntenic gene pairs between cucumber and potato than between *Arabidopsis* and cucumber. *CsCML9* and *CsCML24* were derived from the same gene in *Arabidopsis,* and *CsCML9* and *CsCML43* were associated with two syntenic genes in tomato. Moreover, some *CsCML* genes were not syntenic gene pairs in *Arabidopsis* or tomato, illustrating that these genes might be unique to cucumber.
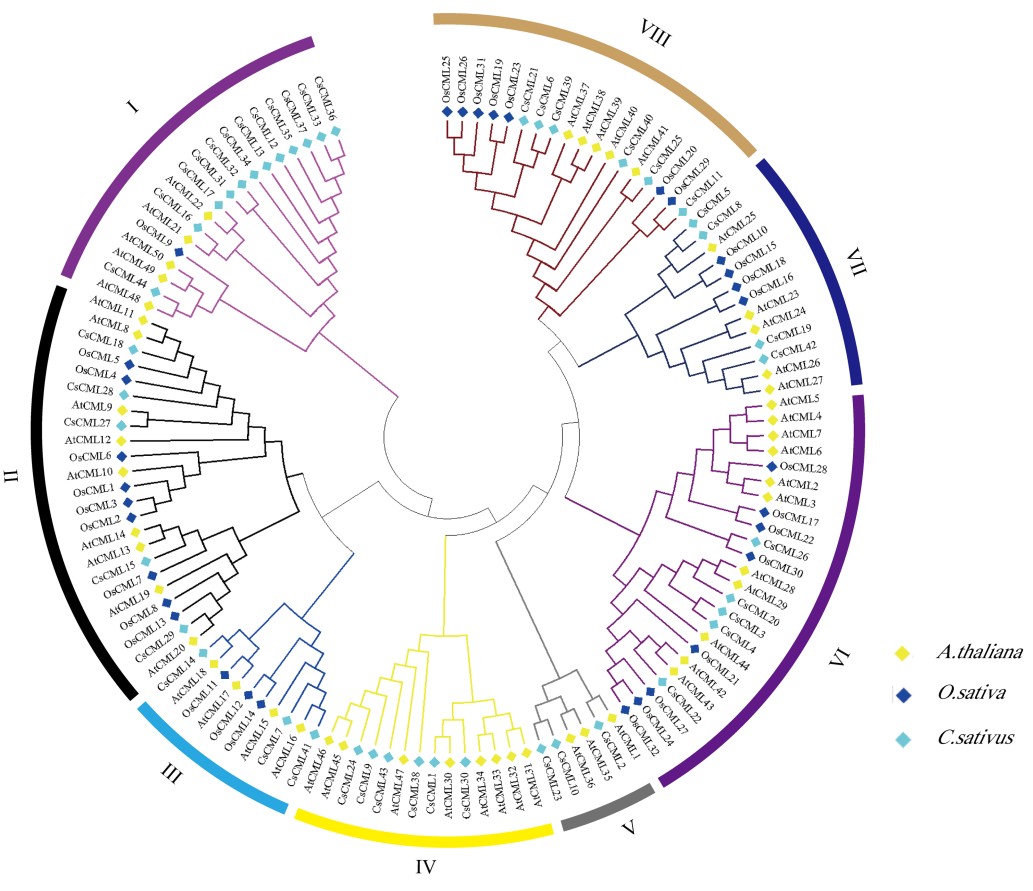

**Figure 3  Phylogenetic tree of CML proteins in *cucum is sativus, Arabidopsis thaliana, Oryza sativa* based on the neighborhood-joining method.** Different colors represent different groups.

## Expression patterns of *CsCML* genes in different tissues and under different conditions

To identify the potential physiological role of *CsCML*s, the expression patterns of 44 *CsCML* genes were investigated by qRT-PCR in the leaves, stems, flowers, peels, and under different stress conditions or different hormone treatments (Data S3), and a heatmap was used to represent the results (Figs. 5A and 5B). The results presented a distinct expression pattern in different organs and conditions.

The *CsCML* genes showed relatively high expression mainly in the flowers, especially *CsCML31*, *CsML3*, and *CsML13*. Several *CsCML* genes were highly expressed in the cucumber peel tissue, such as *CsCML 23, 32, 38*, and *44*. *CsCML28* and *CsCML39* were highly expressed in the leaf, while *CsCML27* and *CsCML36* were strongly expressed in the stem (Fig. 5A). The gene expression differences demonstrated that gene expression specificity exists in different tissues and the *CsCMLs* may be more involved in floral organ morphogenesis.

Plants are subject to a variety of environmental conditions during growth and development. In this study, 44 CsCML displayed disparate expression patterns under

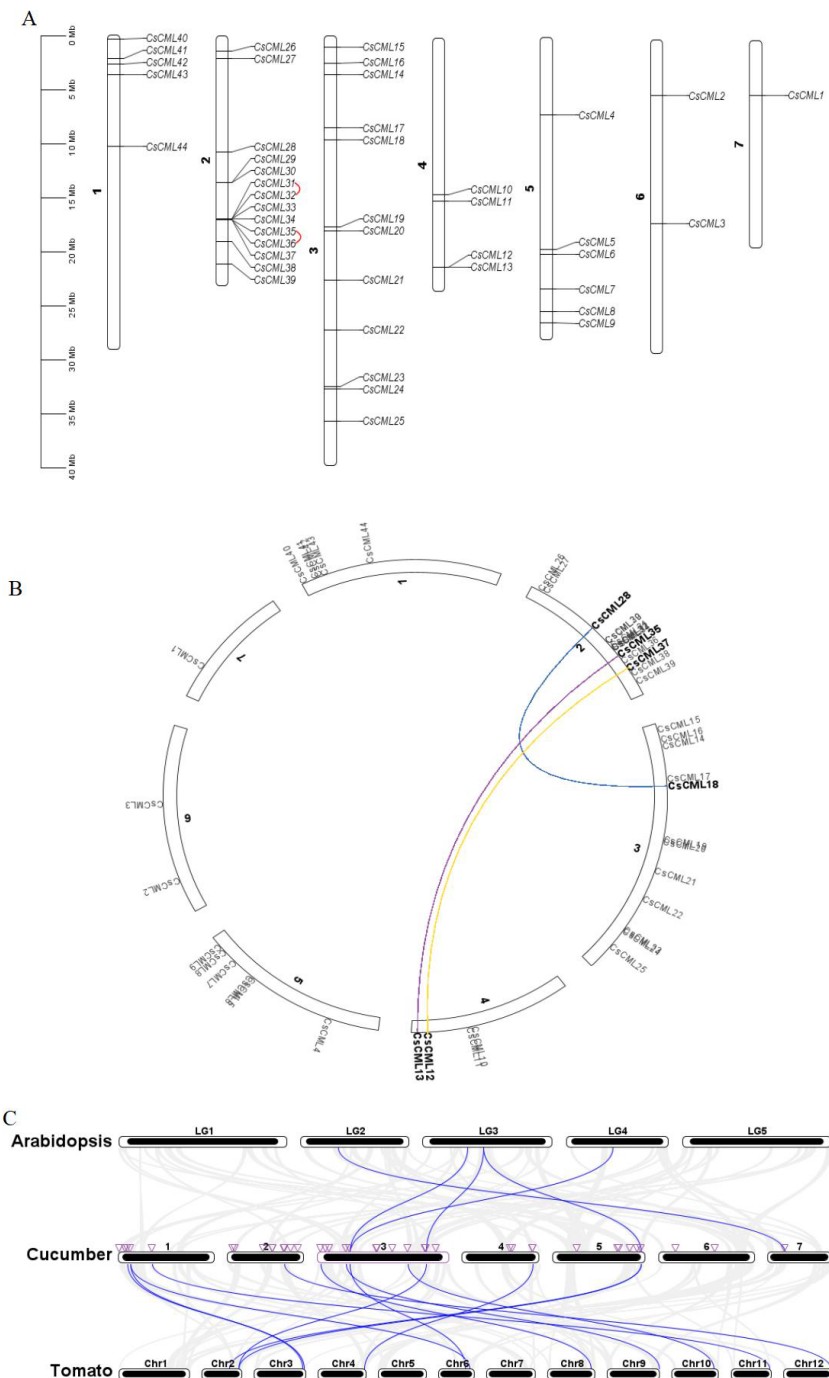

**Figure 4 Chromosome distribution, gene duplication, and synteny of *CsCMLs*.** (A) Chromosome distribution and *CsCML* gene tandem duplication. The red lines indicate tandemly duplicated gene pairs. (B) Interchromosomal relationships. Different colored lines represent the different segmentally duplicated gene pairs of *CsCML*. (C) Synteny analysis of *CsCML* in cucumber, Arabidopsis, and tomato. The blue lines represent the synteny of the *CML* gene in cucumber, *Arabidopsis,* and tomato. The purple triangles represent *CsCML* genes.

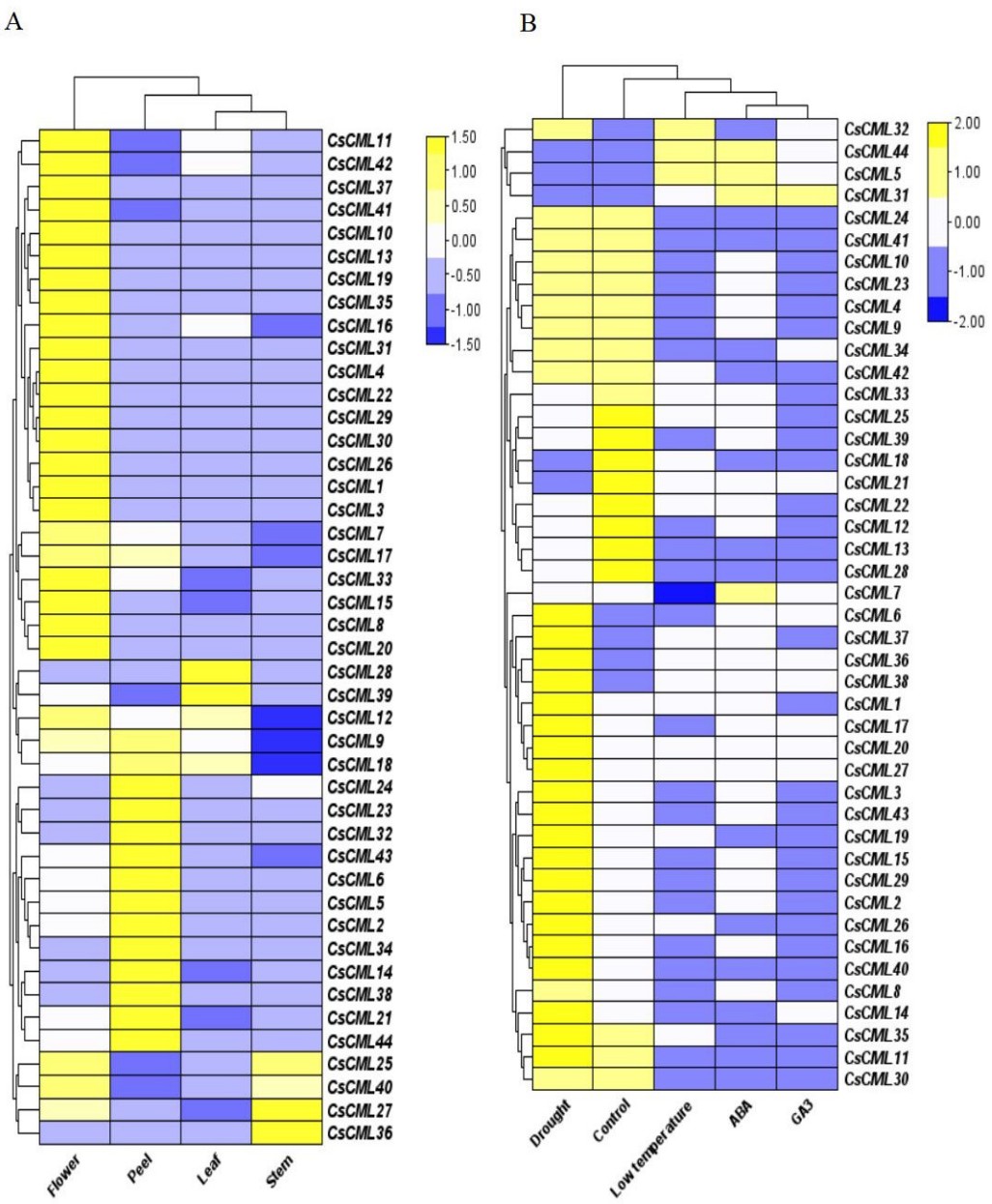

**Figure 5** **The relative expression patterns of *CsCML* genes in different tissues of cucumber under different conditions.** Heatmap showing the relative expression patterns of *CsCML* genes in different tissues of cucumber under different conditions. The heatmap was constructed based on the relative expression of *CsCML* genes determined by qRT-PCR in various tissues (A) and under different conditions (B) and was performed using TBtools. The relative expression was log2 transformed. Each value represents the mean of the relative expression of three replicates. Genes highly or weakly expressed are colored yellow to blue, respectively.

different conditions. Thirty-one *CsCML* genes were highly up-regulated under drought, while *CsCML5*, *CsCML21*, *CsCML31*, *CsCML44*, and *CsCML18* were down-regulated. While the expression pattern under ABA, GA₃, and low temperature showed the similar

trend. In Fig. 5B, one gene (*CsCML31*) was strongly induced by GA$_3$, which was also induced by ABA. Several *CsCML* genes remarkably up- regulated under ABA and low temperature, such as *CsCML5* and *CsCML44*. In addition, *CsCML32* was simultaneously up-regulated under low temperature. These results indicated that *CsCML* genes might play a pivotal role in hormone signal transduction and the response to biotic stress.

### The protein interaction network for CsCMLs

In this study, 44 CsCML proteins were subjected to STRING to predict the protein interaction network in cucumber. Whereas 21 CsCML proteins were involved in the interaction network, and eight CML proteins correlated with more than four other CML proteins, AT1G18210, CML38, and AT2G27480 were associated with nine CsCML proteins. As shown in Fig. 6, CsCML40, CsCML6, CsCML41, and CsCML30 formed the close interaction and represented hypothetic co-occurrence and co-expression. The analysis of the protein interaction network indicated that CsCML regulated the expression of downstream genes by interacting with other proteins and provided a useful resource for further research.

## DISCUSSION

### Identification and characterization of CMLs in cucumber

Forty-four CsCML members were identified in cucumber using 50 *Arabidopsis* CML proteins as queries, which is smaller than tomato (*Munir et al., 2016a*; *Munir et al., 2016b*), wheat (*Liu et al., 2022*), papaya (*Ding et al., 2018*), and ginkgo (*Zhang et al., 2022*). The number of CMLs in cucumber is lower than that in other species, which is likely due to the low number of gene duplications in the cucumber genome (*Asano et al., 2012*); as exhibited by our results (Figs. 4B and 4C). Our bioinformatic analysis indicated that the CsCML molecular weight ranged from 9.42 KDa to 28.31 KDa, and most CsCML proteins tended to be acidic and hydrophilic (Table 1), which was consistent with alfalfa, whose molecular weight varied from 7.37 KDa to 29.98 KDa, and most of MtCMLs were acid (*Sun, Yu & Guo, 2020*). The previous study pointed out that CaM shared a high conserve with CML (*Snedden et al., 2015*). In the *Arabidopsis* (*McCormack & Braam, 2003*) and papaya (*Ding et al., 2018*), the CML shared the 16.1% to 74.5%, 22.4% to 88.1% identity with AtCaM2, respectively. To ensure our prediction accuracy, we adopted the amino acid identity with 16–80% as the selection criterion. At present study, the 44 CsCML shared 24% to 77% amino acid identity with AtCaM2.

Protein post-translational modifications have diverse biological functions related to signal transduction, protein transport, protein regulation, protein localization, and extracellular communication (*Mohanta, Kumar & Bae, 2017*; *Shi & Du, 2020*; *Xu et al., 2015*). Myristoylation and palmitoylation are two major posttranslational modifications. Studies have revealed that proteins possessing myristoylation motifs tend to be in the plasma membrane (*Mehlmer et al., 2010*). Our study indicated that a few *CML* genes (*CsCML3, 15, 16, 17, 19, 21, 28, 30, 32,* and *34*) contained myristoylation and/or palmitoylation sites, which may cause CML conformation changes and promote protein–membrane and protein–protein interactions. CMLs were reported localized in various parts; in this study,

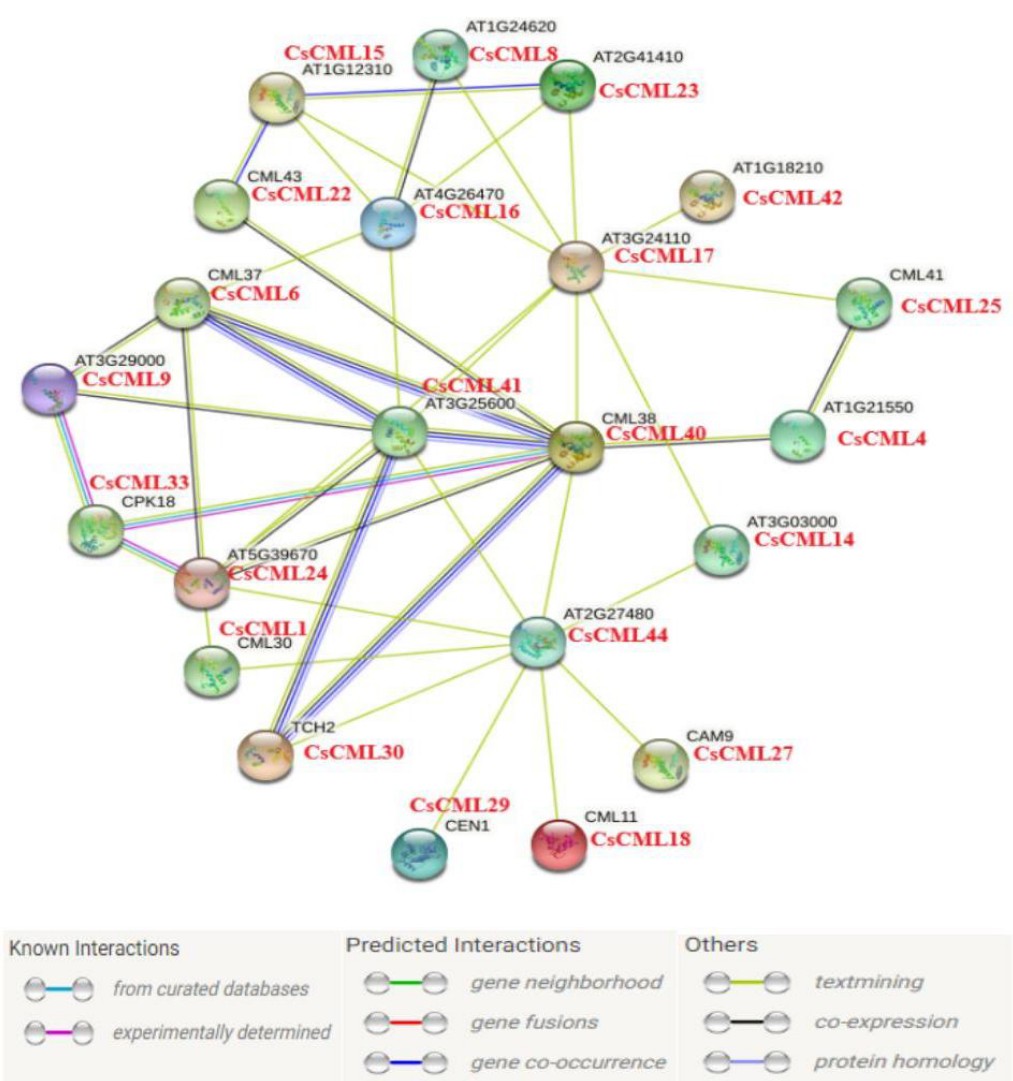

**Figure 6** **Protein interaction network of CML proteins.** The homologous genes from cucumber and *Arabidopsis* are in red and black, respectively.

the CsCML proteins were mainly localized in the nucleus and cytoplasmic, which was in consonance with lotus (*Gao et al., 2022*). CMLs mediate $Ca^{2+}$ signal transduction by binding with $Ca^{2+}$ to form CML/$Ca^{2+}$ complex compounds. CML proteins also contain an EF-hand conserved domain, which can bind to $Ca^{2+}$. Therefore, the number of EF-hands may affect the role of CMLs. In *Arabidopsis,* there are typically two to six EF-hands (*McCormack & Braam, 2003*), while cucumber possesses one to four. This may be the result of differences in the sequence and structure of the CMLs. Previous research found that most *CML* genes are intron-less, while some contain no more than nine introns (*Ding et al., 2018*; *Mohanta, Kumar & Bae, 2017*; *Sun, Yu & Guo, 2020*). Our results showed that most of the cucumber *CMLs* are intron-less, while others have fewer than six introns (Fig. 1). These

introns may have evolved with CaMs, thus possibly exerting positive pressure for *CML* evolution (*Mohanta, Kumar & Bae, 2017*). These results are similar to those reported for *Medicago truncatula* Gaertn. (*Sun, Yu & Guo, 2020*). Besides, the variance of the conserved motif in the CsCML (Fig. 1A) will contribute to the function divergence (*Li et al., 2019a*; *Li et al., 2019b*).

The CsCMLs were divided into eight subgroups (Fig. 3), which is similar to that in apple (*Li et al., 2019a*; *Li et al., 2019b*) and *Medicago truncatula* (*Sun, Yu & Guo, 2020*), while four subgroups are present in wild tomato (*Shi & Du, 2020*) and seven are present in Chinese cabbage (*Nie, Zhang & Zhang, 2017*). These results demonstrated that CMLs are highly conserved among species and perform the similar function. Gene location analysis showed that 44 *CsCML* genes were not uniformly distributed on the seven chromosomes (Fig. 4A). Of the *CsCML* genes, 14 and 12 were on chromosomes 2 and 3, respectively. Gene duplication is considered to play an important role in gene expansion (*Kong et al., 2007*). In our study, three segmentally duplicated gene pairs and two tandemly duplicated gene pairs were found in the cucumber *CsCMLs* (Figs. 4A and 4B). And many *CsCML* genes showed an extensive synteny relationship with *Arabidopsis* and tomato (Fig. 4C). These results were subjected to less duplication in cucumber during evolution, which explains why the *CML* genes in cucumber are less numerous than in *Arabidopsis*. Moreover, this may be due to the long-term adaptation of species to different growth environments (*Guo, 2013*).

## Expression patterns of *CsCML* genes

It has been widely reported that *CMLs* play an important role in plant development and stress responses (*McCormack, Tsai & Braam, 2005*; *McCormack & Braam, 2003*; *Reddy et al., 2011*). In our study, the spatio-temporal expression pattern demonstrated a distinct tissue specificity (Fig. 5A). *CsCML27* and *CsCML36* were only strongly expressed in the stem. Likewise, *CsCML28* and *CsCML39* were only strongly expressed in leaves. As paralogous genes may perform similar functions, the evolutionary relationships and potential functions of the *CMLs* were explored in a phylogenetic tree of cucumber and *Arabidopsis* (Fig. 3). Microarray data showed that *AtCML21* had pollen-specific expression (*Becker et al., 2003*). In addition, *AtCML15* was depicted as involving floral development (*Ogunrinde et al., 2017*). Our results indicated that the homologous genes *CsCML41* and *CsCML7* were highly expressed in the flowers (Fig. 5A). Moreover, *CsCML13* and *CsCML31* were strongly upregulated in the flowers, which are in the same subgroup with *AtCML21* (Fig. 3), indicating that they may participate in flowering and fruit growth. This result corroborates findings in *Arabidopsis* (*McCormack, Tsai & Braam, 2005*).

CMLs play a vital role in response to biotic and abiotic stress. In *Arabidopsis*, *CML8* (*Zhu et al., 2017*) and *CML9* (*Leba et al., 2012*) was strongly induce by *Pseudomonas syringae*. Besides, *AtCML9* involved in salt tolerancce through its effects on the ABA-mediated pathways (*Magnan et al., 2008*). One study also reported that *AtCML37* was positively regulated by drought, while *AtCML42* exhibited opposite function (*Heyer et al., 2022*). Additionally, the overexpression of *ShCML44* enhanced tolerance during cold and drought (*Munir et al., 2016a*; *Munir et al., 2016b*).

In the present study, *CsCML5*, *32*, and *44* were strongly induced under low temperature stress, and high numbers of *CsCMLs* such as *CsCML1*, *6*, *17*, and *20* were highly expressed under drought(Fig. 5B). As cucumber is a cold-sensitive vegetable, these genes might be good candidates for stress tolerance. It has been reported that *CML* genes are induced inordinately by different hormones (*McCormack, Tsai & Braam, 2005*; *Midhat et al., 2018*). Some *CsCML* genes exhibited similar expression patterns under ABA and GA$_3$ treatment; for example, *CsCML31* was up-regulated under both ABA and GA$_3$, whereas, *CsCML41* was down-regulated (Fig. 5B), suggesting that these genes may be commonly involved in response to ABA and GA. Moreover, Fig. 6 showed the protein interaction network in *Arabidopsis* and cucumber, deduced that *CsCML6*, *CsCML30*, *CsCML40,* and *CsCML41* were co-expressed to participate in cucumber response to drought. The *CsCML* genes in this study were distinctively expressed in different tissues and induced and/or suppressed by different hormones and stressors. These results were consistent with previous findings that the CML genes are involved in hormone and other stress responses because they contains *cis*-acting elements.

## CONCLUSIONS

Forty-four *CsCML* genes were identified from the cucumber genome. Gene structure and sequence analysis showed that these *CsCML* genes containing one to four highly-conserved EF-hand functional domains were unevenly located on seven chromosomes. *Cis*-acting element analysis indicated that these genes might respond to multiple hormones and stresses. Spatiotemporal expression analysis results confirmed that *CsCML* genes play a vital role during plant development and stress resistance. Altogether, this study provides a good foundation for further studies of the functions of *CsCML* genes in cucumber.

## ACKNOWLEDGEMENTS

We thank LetPub for its linguistic assistance during the preparation of this manuscript.

### Funding

This work was supported by funding from the Natural Science Foundation of Guangxi Province (2020GXNSFAA297007, 2020GXNSFAA297153 and 2019GXNSFBA245037) and the Doctoral Start-up Fund of Hezhou University (HZUB202008). The funders had no role in study design, data collection and analysis, decision to publish, or preparation of the manuscript.

### Grant Disclosures

The following grant information was disclosed by the authors:
The Natural Science Foundation of Guangxi Province: 2020GXNSFAA297007, 2020GXNSFAA297153, 2019GXNSFBA245037.
The Doctoral Start-up Fund of Hezhou University: HZUB202008.

## Competing Interests

The authors declare there are no competing interests.

## Author Contributions

- Yunfen Liu conceived and designed the experiments, performed the experiments, prepared figures and/or tables, authored or reviewed drafts of the article, and approved the final draft.
- Feilong Yin performed the experiments, analyzed the data, prepared figures and/or tables, and approved the final draft.
- Lingyan Liao performed the experiments, analyzed the data, prepared figures and/or tables, and approved the final draft.
- Liang Shuai conceived and designed the experiments, authored or reviewed drafts of the article, and approved the final draft.

## DNA Deposition

The following information was supplied regarding the deposition of DNA sequences:

The OsCML sequences are available at the Rice genome annotation project: LOC_Os01g59530 to LOC_Os08g04890.

Available at http://rice.uga.edu/cgi-bin/gbrowse/rice/

The AtCML and CsCML sequences are available at EnsemblePlants: AT3G59450 to Csa_1G164670.

Available at https://plants.ensembl.org/index.html

## Data Availability

The raw data are available in the Supplemental Files.

## Supplemental Information

Supplemental information for this article can be found online at http://dx.doi.org/10.7717/peerj.14637#supplemental-information.

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
