# Peer review of "Genome-wide identification and expression analysis of calmodulin-like proteins in cucumber"

_PeerJ, doi:10.7717/peerj.14637_

## Round 0.1 · original submission · Major Revisions

Revise the manuscript as per the reviewer's comments.

·

Basic reporting

In this study, the authors investigated the genome-wide identification of calmodulin-like proteins in cucumber and their expression profiles in response to hormonal and abiotic stresses. The paper is the most common analysis of gene family identification. This paper still needs to be revised before it can be published.

Experimental design

1.The paper is a very common gene family identification and expression analysis, the content and form of the paper is not particularly innovative, whether the authors can explore the other characteristics of the class of proteins to add in the paper?

2.The statistical analysis is missing in materials and methods section, please check and add it.

Validity of the findings

1.In order to make the result of Phylogenetic relationship, gene structure, conserved protein motifs, and putative cisacting elements in CsCMLs clearer and more intuitive, I suggest that the author readjust Figure 1 to make it as clear as possible.

2.In the data analysis section of Figure 5A, the authors say that defining the relative expression of leaves as 1 is inherently problematic, and published studies suggest that using the median of the overall expression as a benchmark is the most appropriate way to process tissue-specific expression data.

3.In stress and hormonal treatment, why compare expression in leaves and not to untreated? There is a big problem in the data analysis section, please explain the reason and make changes.

Additional comments

The note in Table 1 need to be supplemented with what does the "+" stands for.

Reviewer 2 ·

Basic reporting

In this study, 44 CsCMLs family members were characterized, as well as some bioinformatics analysis were conducted. The study establishes a foundation for further function studies of CsCMLs. However, I think the manuscript was not innovative and rigorous enough for publication now. The majors was:
1. Calmodulin share high identity with calmodulin-like protein, as well as CaM and CML all just possess EF-hand motifs without any other functional domain, how to exclude the calmodulin gene from Blast results?
2. All software used in this study should specify the parameters.
3. Some writing should be more accurate, for example:
Line 168: “The CsCML proteins contained two to four EF-hand domains, except CsCML4, which possessed only one.”, should be “Most of CsCML proteins contained two to four EF-hand domains, except CsCML4, which possessed only one.”
Line 188:“Exon-intron analysis showed that most of the CsCML members had no introns, while others had one to six introns”, it is better to state clearly how many CsCML members had no introns, and how many members had one to six introns.

Experimental design

no comment

Validity of the findings

no comment

Additional comments

no comment

Reviewer 3 ·

Basic reporting

The authors present the search and characterization of Calmodulin family members in Cucumber sativus, using as input the existing reference proteome for the species. The article is well written, but its introduction should be updated and the discussion should be expanded, contextualizing with other works in the current literature. I attach my comments
1) The authors should do an updated review of the literature. All the papers cited in the introduction (with the exception of 2) are pre-2020. Searching Google Scholar for the keywords in this paper yields hundreds of post-2020 results, indicating that there has been a major update of the literature in this regard, which should be considered in this paper.
2) On line 93, add the citation corresponding to Ensembl Plants (Yates et al 2022, https://doi.org/10.1093/nar/gkab1007)
3) Search and annotation using BlastP, followed by analysis by PFAM, is a perfectible strategy, as they may be missing CML. In these strategies, it is advisable to do one of a larger number of databases. A free and complete tool is Interproscan, which allows in a single run to apply more than 14 databases of motifs and domains (i.e. interpro, Panther, pfam, gene3D, among others). I recommend repeating the analysis, but using this larger compendium of databases.
4) The authors mention that they discarded sequences without EF-hand predicted. But many times PFAM predicts domains, even if they are incomplete. What criteria did the authors applied to determine which of the predicted ones to stay with? only those with the complete domain? only those with at least a certain % of the domain? If the latter, what was that percentage? Clarify in MyM.
5) In the discussion, the manuscript needs to "dialogue" with other works in the current literature, and discuss or put in context the results. For example, in the section "Genome-wide identification and characterization of CML in cucumber", where they study IP, subcellular localization, myristoylation and palmitoylation sites, among others, the authors do not discuss anything. Was anything similar seen in CML studies in other species? How are these results interpreted, in terms of this study?

Experimental design

all my comments were previously detailed

Validity of the findings

all my comments were previously detailed

Additional comments

all my comments were previously detailed

---

## Round 0.2 · accepted · Accept

All the reviewer comments are satisfactorily addressed.

·

Basic reporting

The authors have fully revised the issues raised last time in the revision. I agree to accept this paper for publication.

Experimental design

Please see the comments for "1. Basic reporting".

Validity of the findings

Please see the comments for "1. Basic reporting".

Additional comments

Please see the comments for "1. Basic reporting".

Reviewer 2 ·

Basic reporting

I think the authors have studied the comments carefully and have made correction which meet with approval.

Experimental design

no comment

Validity of the findings

no comment